# IL6-mediated HCoV-host interactome regulatory network and GO/Pathway enrichment analysis

**Gianfranco Politano**, **Alfredo Benso***

Computer Science and Automation Department, Politecnico di Torino, Italy

* alfredo.benso@polito.it

**Data Availability Statement:** All relevant data are within the manuscript and its Supporting Information files.

**Funding:** The authors received no specific funding for this work.

## Abstract

During these days of global emergency for the COVID-19 disease outbreak, there is an urgency to share reliable information able to help worldwide life scientists to get better insights and make sense of the large amount of data currently available. In this study we used the results presented in [1] to perform two different Systems Biology analyses on the HCoV-host interactome. In the first one, we reconstructed the interactome of the HCoV-host proteins, integrating it with highly reliable miRNA and drug interactions information. We then added the IL-6 gene, identified in recent publications [2] as heavily involved in the COVID-19 progression and, interestingly, we identified several interactions with the reconstructed interactome. In the second analysis, we performed a Gene Ontology and a Pathways enrichment analysis on the full set of the HCoV-host interactome proteins and on the ones belonging to a significantly dense cluster of interacting proteins identified in the first analysis. Results of the two analyses provide a compact but comprehensive glance on some of the current state-of-the-art regulations, GO, and pathways involved in the HCoV-host interactome, and that could support all scientists currently focusing on SARS-CoV-2 research.

## Author summary

In this paper we provide data about the HCoV-host interactome that can be extracted from the integration of several public available databases. We used the initial interactome published by Zhou et al. and analyzed if there are already known and validated interactions. We also looked into possible known miRNAs and drugs interactions to suggest possible biomarker candidates and treatment options. We also performed a Gene Ontology and a Pathways enrichment analysis to understand which are the pathways most likely involved in the proteins targeted by SARS-CoV-2. This paper not only provides a set of validated and reliable data that could help researchers in their fight against the COVID-19 disease outbreak, but also demonstrates how Systems Biology can be effectively used to quickly gather preliminary but still significant data without resorting only to expensive lab experiments.

**Competing interests:** The authors have declared that no competing interests exist.

## Introduction

The sudden global emergency caused by the newly discovered SARS-CoV-2 requires a fast but reliable and comprehensive analysis of the virus interactions with the human genome. Corona-viruses (CoVs) typically affect the respiratory tract of mammals, including humans, and lead to mild to severe respiratory tract infections [3]. To provide fast results that could contribute to the fight against the epidemic, in this study we started from the results presented in [1] and performed two different Systems Biology analyses on the HCoV-host interactome. In the first one, relying on validated interaction only, we reconstructed the regulatory network linking the proteins identified in [1], integrating it with the IL-6 protein, miRNA, and drug interactions information. We would like to stress the fact that, while the original paper [1] focuses on Protein Interactions, this paper looks at gene regulatory mechanisms. We wanted to look at the same problem, starting from Zhou results, from a different point of view. We consider this contribution as a continuation or expansion of Zhou work, not something that in any way contradicts it.

Our first analysis has been done using the RING database [4], a data repository integrating 30 public databases designed for advanced biological networks reconstruction. This phase allowed to immediately identify a strongly connected cluster of proteins and drugs, as well as 3 miRNAs that the cluster produces. In the second analysis, executed using an R pipeline, we compared the full set of the HCoV-host interactome proteins with the ones belonging to the identified cluster. In particular, we performed a Gene Ontology [5] enrichment analysis, a Pathways enrichment analysis (with both KEGG [6] and WikiPathways [7]), as well as a final cluster analysis. The results of the latter analysis are publicly available at https://precious. polito.it/covid-19/ as an interactive website (the same data is also available in the S1_File of the supplemental materials).

As pure bioinformaticians we currently lack a valid experimental setup to prove the validity of results and the goal of this paper is not to propose a new methodology but to apply consoli-dated techniques to share and propagate knowledge that may help other scientists involved in SARS-CoV-2 research to better design their studies or uncover new hypothesis.

## Methods and results

During these days of global emergency for the COVID-19 disease outbreak, there is an urgency to share reliable information able to help worldwide life scientists to get better insights and make sense of the large amount of data currently available. Based on these premise, we aim at providing at-a-glance insights, easy to read by life scientists, about: i) the current state-of-the-art knowledge available in terms of direct regulatory interactions taking place among gene/proteins included in the HCoV-host interactome [1] and IL-6, which resulted in the HCoV-Cluster net-work, and ii) the whole set of GOs and Pathways enriched in both the HCoV sets.

### Network analysis

For this first analysis we used the Graph Tools of the RING database (https://precious.polito. it/theringdb/login) [4]. This tool integrates more than 30 publicly available data repositories and allows to reconstruct interaction networks between genes, Transcription Factors, miR-NAs, Drugs, Diseases, and SNPs.

As a first step we performed a network reconstruction looking only for interactions among the proteins of the HCoV-host interactome as reported in S3 Table in the Supplemental mate-rial of [1]. In order to build the minimum set of most reliable interactions, we used the most conservative settings, where all interactions are validated, manually curated, and **limited to**

**signaling or regulatory actions** like inhibition/activation (data sources: TRRUST [8] and SIGNOR [9]). This phase allowed to divide the original HCoV-host interactome proteins in two sets: one that shows no obvious regulatory interactions, and one made of 20 genes (see Table 1) out of the original 135 proteins, that forms a well-defined regulatory interactome. For all interactions we report (in S3 Table of the supplemental material), the Pubmed references of the supporting papers.

The HCoV-host Network has been then enhanced looking for co-expressed microRNAs (in this case we selected the MIRIAD data source [10], while relaxing the Validated and Manually curated filters). This operation highlighted that 3 miRNAs are co-expressed by the identified cluster: hsa-miR 3912-3p and hsa-miR 3912-5p hosted by the NPM1 gene, and hsa-miR 4751, hosted by ATF5 the gene. We decided to highlight miRna possibly involved in the Covid interactome because miRNAs are known to mediate several regulatory mechanisms, but also to be powerful biomarkers for several diseases. Finally, we added the IL-6 gene to see if it presented any interaction with the identified cluster.

The resulting network is presented in Fig 1. Yellow edges represent multiple interactions. For each node of the network, the S1 Table reports the node name, type, UniprotId, and possible aliases. For each interaction, the S2 Table reports the interaction type (to match the symbol with its meaning refer to [4]), the database of origin, and, where available, the PubMed Ids of the related papers.

As a third step, we further enhanced the network looking for drug interactions that had at least two targets in the identified cluster (data sources: DGIdB [11] and DRUGBANK[12]). In

**Table 1. List of highly-likely interacting genes within the HCoV-host interactome.**

| TYPE | ID | NAME | EXTRA INFO |
|------|-----|------|------------|
| TF | 22809 | ATF5 | UniprotId: A0A024QZG3, Aliases: ATFX\|HMFN0395, Map Location: A0A024QZG3 |
| gene | 596 | BCL2 | UniprotId: A0A024R2B3, Aliases: Bcl-2\|PPP1R50, Map Location: A0A024R2B3 |
| gene | 598 | BCL2L1 | UniprotId: A0A0S2Z3C5, Aliases: BCL-XL/S\|BCL2L\|BCLX\|Bcl-X\|PPP1R52, Map Location: A0A0S2Z3C5 |
| gene | 2931 | GSK3A | UniprotId: A0A024R0L5, Aliases: -, Map Location: A0A024R0L5 |
| TF | 2932 | GSK3B | UniprotId: P49841, Aliases: -, Map Location: P49841 |
| gene | 3569 | IL-6 | UniprotId: B4DNQ5, Aliases: BSF-2\|BSF2\|CDF\|HGF\|HSF\|IFN-beta-2\|IFNB2\|IL-6, Map Location: B4DNQ5 |
| TF | 3725 | JUN | UniprotId: P05412, Aliases: AP-1\|AP1\|c-Jun\|p39, Map Location: P05412 |
| gene | 3837 | KPNB1 | UniprotId: B7Z752, Aliases: IMB1\|IPO1\|IPOB\|Impnb\|NTF97, Map Location: B7Z752 |
| gene | 4170 | MCL1 | UniprotId: A0A087WT64, Aliases: BCL2L3\|EAT\|MCL1-ES\|MCL1L\|MCL1S\|Mcl-1\|TM\|bcl2-L-3\|mcl1/EAT, Map Location: A0A087WT64 |
| gene | 4666 | NACA | UniprotId: A0A024RB41, Aliases: HSD48\|NAC-alpha\|NACA1\|skNAC, Map Location: A0A024RB41 |
| TF | 4869 | NPM1 | UniprotId: A0A0S2Z491, Aliases: B23\|NPM, Map Location: A0A0S2Z491 |
| TF | 142 | PARP1 | UniprotId: A0A024R3T8, Aliases: ADPRT\|ADPRT 1\|ADPRT1\|ARTD1\|PARP\|PARP-1\|PPOL\|pADPRT-1, Map Location: A0A024R3T8 |
| TF | 5245 | PHB | UniprotId: A8K401, Aliases: HEL-215\|HEL-S-54e\|PHB1, Map Location: A8K401 |
| gene | 5499 | PPP1CA | UniprotId: A0A140VJS9, Aliases: PP-1A\|PP1A\|PP1alpha\|PPP1A, Map Location: A0A140VJS9 |
| gene | 6502 | SKP2 | UniprotId: A0A024R069, Aliases: FBL1\|FBXL1\|FLB1\|p45, Map Location: A0A024R069 |
| TF | 4088 | SMAD3 | UniprotId: A0A024R5Z3, Aliases: HSPC193\|HsT17436\|JV15-2\|LDS1C\|LDS3\|MADH3, Map Location: A0A024R5Z3 |
| TF | 6774 | STAT3 | UniprotId: P40763, Aliases: ADMIO\|ADMIO1\|APRF\|HIES, Map Location: P40763 |
| TF | 6776 | STAT5A | UniprotId: A8K6I5, Aliases: MGF\|STAT5, Map Location: A8K6I5 |
| TF | 7040 | TGFB1 | UniprotId: P01137, Aliases: CED\|DPD1\|LAP\|TGFB\|TGFbeta, Map Location: P01137 |

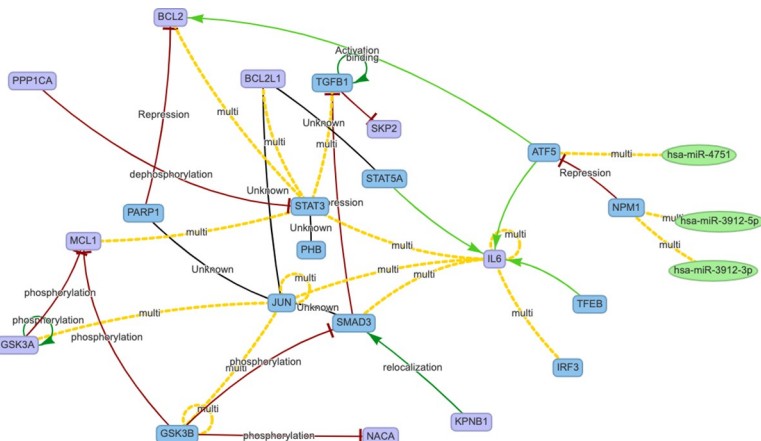

**Fig 1. HCoV-host and IL-6 interactome cluster.** Nodes are represented as follow: genes in violet, transcription factors in blue, and microRNAs in green. Edges are marked in: red for inhibition/silencing effects, green for activation/silencing, black for undirected/unknown and yellow when multiple regulations are available in literature (see the list of all the interactions reported in Supplemental Material in S2 Table).

this way we limited the number of drugs to only the ones that possibly have a stronger effect on the HCoV-Cluster of proteins, by presenting multiple targets. The resulting network is presented in Fig 2.

For the sake of clarity and readability we kept the proposed HCoV interacting network as small as possible and we avoided including further possible regulations with weaker reliability. Custom enhancements are obviously possible for interested scientists, and instructions are reported in the S1 Instruction file.

The list of drugs that have at least two interactions with the cluster is reported in Table 2. Although, the list of drugs reported only shares Paroxetine with of [1], the others could still be of interest as possible repurposing candidates.

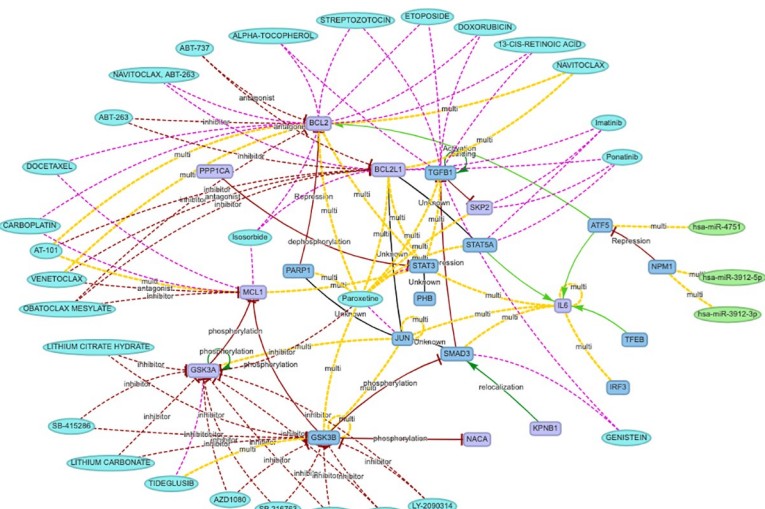

**Fig 2. HCoV-host and IL-6 interactome cluster enriched with drugs interaction data.** Nodes are represented as follow: genes in violet, transcription factors in blue, microRNAs in green, and drugs in cyan. Edges are marked in: red for inhibition/silencing effects, green for activation/silencing, black for undirected/unknown, fuchsia for drug-target association, and yellow when multiple regulations are available in literature (see the list of all the interactions reported.

**Table 2. List of possible drug interactions with the identified HCoV-host cluster.**

| Name | # targets | DB reference |
|---|---|---|
| Paroxetine | 10 | PubchemId: 43815, PharmGKB: PA450801, DrugBank: DB00715 |
| Azd-1080 | 4 | |
| Isosorbide | 3 | PubchemId: 12597, DrugBank: DB09401 |
| Imatinib | 3 | PubchemId: 5291, PharmGKB: PA10804, DrugBank: DB00619 |
| Ponatinib | 3 | PubchemId: 24826799, PharmGKB: PA165980594, DrugBank: DB08901 |
| Venetoclax | 3 | PubchemId: 49846579, PharmGKB: PA166153473, DrugBank: DB11581 |
| Navitoclax | 2 | PubchemId: 24978538, DrugBank: DB12340 |
| Lithium carbonate | 2 | PubchemId: 11125 |
| Tideglusib | 2 | PubchemId: 11313622, DrugBank: DB12129 |
| Docetaxel | 2 | PubchemId: 148124, PharmGKB: PA449383, DrugBank: DB01248 |
| Carboplatin | 2 | |
| Genistein | 2 | PubchemId: 5280961, PharmGKB: PA165109660, DrugBank: DB01645 |
| Streptozotocin | 2 | PubchemId: 2733335 |
| Lithium citrate hydrate | 2 | PubchemId: 2724118 |
| Obatoclax mesylate | 3 | |
| Ly-2090314 | 2 | PubchemId: 10029385, DrugBank: DB11913 |
| Chir-99021 | 2 | PubchemId: 9956119 |
| Doxorubicin | 2 | PubchemId: 31703, PharmGKB: PA449412, DrugBank: DB00997 |
| Etoposide | 2 | PubchemId: 36462, PharmGKB: PA449552, DrugBank: DB00773 |

The proposed HCoV interacting network may be helpful to all scientists currently involved in SARS-CoV-2 research, since it provides a compact but comprehensive glance on some of the current state-of-the-art regulations that take place among the HCoV-host interactome.

## Enrichment analysis

For this second analysis we resorted to an R [13] pipeline, taking advantage of *clusterProfiler* package [14], and we fed it with both the full-set of 135 HCoV-host proteins, and HCoV cluster of 20 proteins identified in the previous analysis.

With both the sets we performed statistically valid enrichment analysis on:

- the three Gene Ontologies (GO [5]) (i.e., Molecular Function, Cellular Component and Biological Process);

- two of the most widely used pathway repositories KEGG [6] and WikiPathways[7]);

Finally, we also added a one non-statistical GO classification.

The result is provided as an interactive website (available at https://precious.polito.it/covid-19) that collects and presents all results both in graphical and tabular form; all the tables are immediately available to download to, once again, provide scientists with immediately usable result for further analysis. All results are also available in the S1 Data file.

Each enrichment analysis result is provided with its entity (i.e., a GO or Pathway) defined with its ID and Description, the gene ratio coverage, a list of proteins involved in that pathway or GO, and its enrichment p-value adjusted by Benjamini & Hochberg (BH) method [14]. This value has been computed to better control the expected proportion of false discoveries amongst the rejected hypotheses (i.e., false discovery rate, FDR), thus allowing for a less stringent condition on false discoveries, and allowing more candidate options to be included as results. Furthermore, we also computed, only for GOs, a non-statistical classification, in this

case, possibly helpful for preliminary evaluation or hypothesis definition. In this case we simply showcase the frequency of recurring GOs among the entities, with no statistical consideration. Because of that, data tables related to GO classification do not report any p-value.

Finally, a cluster analysis has been computed in order to selectively elucidate possible inner differences between the HCoV-host protein-set and the HCoV-host proteins cluster-set in all the computations performed in the previous steps (i.e., GO Classification GO Enrichment and Pathway Enrichment). The same statistical assumption and limits, previously discussed for enrichment analysis, do apply to cluster analysis.

As a proof of concept, we hereby show in Table 3, the top five Wiki Pathway enriched pathways.

WP3872 describes integrin mediated cell survival regulation induced by parathyroid hormone-related protein. Although the pathway may seem not of immediate applicability for SARS-CoV-2, it is interesting to highlight the involvement of the PI3-K/Akt pathway by increasing levels of integrin A6B4, which further modulate the pro/anti-apoptosis members in the Bcl-2 family. Bcl-2 family of genes have been shown to play an important role in the IL-6–mediated protective response to oxidative stress. Authors in [15] showed that IL-6 induced Bcl-2 expression, both in vivo and in vitro, disrupted interactions between proapoptotic and antiapoptotic factors, and suppressed H2O2-induced loss of mitochondrial membrane potential in vitro. Concluding that IL-6 induces Bcl-2 expression to perform cytoprotective functions in response to oxygen toxicity, and conclude that IL-6 induces Bcl-2 expression to perform cytoprotective functions in response to oxygen toxicity, and that this effect is mediated by alterations in the interactions between BAK and MFNS.

WP4298 refers to Viral Myocarditis (VM) pathway. VM is a rare cardiac disease associated with the inflammation and injury of the myocardium result of cooperation between viral processes and the adaptive as innate host's immune response (see [16–18]). Recent papers address SARS-CoV-2 as responsible for acute myocarditis or fulminant myocarditis nevertheless author state that the mechanism of cardiac pathology caused by SARS-CoV-2 needs further study [19, 20].

WP127 and WP286 are related to interleukine-mediated (IL-5, IL-3) inflammatory response.

Acute Respiratory Distress Syndrome (ARDS) induced by SARS-CoV-2, has been recently highlighted as mediated by high level of cytokine IL-6 [2, 21] that leads to excessive inflammatory response, which is further related to bad prognosis. While IL-6 may be considered as a therapeutic target on his own, common ILs regulatory traits (shared with IL-3 and IL-5, may be helpful to highlight more detailed mechanisms of action in the inflammatory response.

WP3646 pathway reports main hub genes and their related miRNAs. The pathways has been built on a set of differentially expressed genes in both chronic HCV (hepatitis C virus)

**Table 3. List of the top 5 enriched Wiki pathways.** To see the patway full content in the original wikipathway website use the following link https://www.wikipathways.org/index.php/Pathway:<PathwayID> by replacing <PathwayID> with the id reported in the table.

| PathwayID | Name | Gene ratio | BG ratio | Adj. p-value | Genes |
|---|---|---|---|---|---|
| WP3872 | Regulation of Apoptosis by Parathyroid Hormone-related Protein | 7/70 | 22/6249 | 0 | BCL2L1 BCL2L2 MCL1 BCL2A1 BCL2 GSK3A GSK3B |
| WP4298 | Viral Acute Myocarditis | 8/70 | 85/6249 | 0 | STAT3 TGFB1 BCL2L1 BCL2 GSK3B PABPC1 PARP1 CAV1 |
| WP127 | IL-5 Signaling Pathway | 6/70 | 40/6249 | 0 | STAT5A JUN STAT3 BCL2 GSK3A GSK3B |
| WP286 | IL-3 Signaling Pathway | 6/70 | 49/6249 | 0.001 | STAT5A JUN STAT3 TGFB1 BCL2L1 BCL2 |
| WP3646 | Hepatitis C and Hepatocellular Carcinoma | 6/70 | 51/6249 | 0.001 | JUN STAT3 TGFB1 SMAD3 BCL2L1 RRM2 |

and HCC (hepatocellular carcinoma) to highlight how Hepatitis C Virus leads to hepatocellular carcinoma [22]. This pathway suggests a possible similar behavior for corona family viruses and the liver involvement during infection has been recently highlighted, but still largely uncovered [23].

## Supporting information

**S1 Table. List of all nodes.**
(CSV)

**S2 Table. List of all the interaction identified in the cluster network.**
(CSV)

**S3 Table. List of all nodes of the cluster.**
(CSV)

**S1 Data. PDF form of the GO and Pathway enrichment analysis.** Available also as an interactive website at https://precious.polito.it/covid-19/.
(PDF)

**S1 Instruction. Instructions on how to replicate the experiments.**
(PDF)

**S1 File. Network in sif format.**
(SIF)

## Author Contributions

**Conceptualization:** Gianfranco Politano, Alfredo Benso.

**Data curation:** Gianfranco Politano.

**Formal analysis:** Gianfranco Politano.

**Methodology:** Gianfranco Politano, Alfredo Benso.

**Software:** Gianfranco Politano, Alfredo Benso.

**Validation:** Alfredo Benso.

**Writing – original draft:** Gianfranco Politano, Alfredo Benso.

**Writing – review & editing:** Alfredo Benso.

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
