## [Decision Letter · Decision Letter 0]

22 May 2020

Dear Prof. Benso,

Thank you very much for submitting your manuscript "A Systems Biology study of an IL6-mediated HCoV-host interactome: Drug Repurposing and GO/Pathway Enrichment analysis" for consideration at PLOS Computational Biology.

As with all papers reviewed by the journal, your manuscript was reviewed by members of the editorial board and by several independent reviewers. In light of the reviews (below this email), we would like to invite the resubmission of a significantly-revised version that takes into account the reviewers' comments.

We cannot make any decision about publication until we have seen the revised manuscript and your response to the reviewers' comments. Your revised manuscript is also likely to be sent to reviewers for further evaluation.

Sincerely,

James M. Briggs, Ph.D.

Associate Editor

PLOS Computational Biology

William Noble

Deputy Editor

PLOS Computational Biology

Reviewer's Responses to Questions

**Comments to the Authors:**

Reviewer #1: In this paper, the authors performed some preliminary analysis on the HCoV-host interactome.

I am aware that some of the provided results may be useful for the community. As stated by the authors, the computed enrichment analysis may allow scientists to better and more reliably infer candidate pathways and GOs for further inspection.

However, the present paper does not provide a significant improvement in terms of methodologies, nor provides a significant outcome in terms of results. Rather, it may be considered as the result of few queries performed on existing online repositories.

Therefore, although such results may deserve to be shown to the community, since the paper lacks both methodological originality and significance of results, I think that it cannot be published on PLOS Computational Biology in its current form.

Reviewer #2: This paper by politano and benso describes a computational approach to study the HCoV-host interactome. The purpose is to enable scientists with limited experience with compuational analyses to study the HCoV infections and potential drug repurposing/targeting. The paper elegantly shows how many of the results presented in a recent paper by Zhou et al can be achieved by non-programmers using extremely easy-to-use tools previously developed by the authors and adapted to this specific case. These types of workflows are really useful to speed up research in highly time sensitive cases such as the COVID-19 pandemic. The manuscript is well written, and with a few exceptions (listed below) the results are presented in a clear and concise manner.

One major issue that must be resolved before publication can be considered: The link https://precious.polito.itcovid-19/ doesn't work, so it's not really possible to evaluate the feature

Minor comments and suggestions:

It seems to me that the title promises a lot more than what is delivered in this paper. May I suggest a re-wording to more accurately reflect what I see as the ultimate value of this paper: the fact that it enables fast and easy reproduction of extremely advanced computational analyses by scientists with no explicit programming experience.

The first paragraph of materials and methods feels like it belongs in the introduction as it concisely describes the scope and provides essential context for the rest of the work presented.

Figure 1 and 2: Legends seem incomplete. What do the node and edge colors/shapes represent? I would also suggest to include the virus proteins in these graphs.

Figure 2: All of these drugs interact with human proteins, none with virus proteins. What do these drugs do to the proteins they target? They may not affect HCoV-host interaction at all (or they may even enhance it). I realize that it's not within the scope of the paper to dive into the biology, but these results should be discussed a bit more to highlight the value of this feature.

Section 3.2. "three main branches" - they are the only three ontologies in the gene ontology

Reviewer #3: In the manuscript titled “A Systems Biology study of an IL6-mediated HCoV-host interactome: Drug Repurposing and GO/Pathway Enrichment analysis” by Politano and Benso, the authors describe a computational systems biology analysis of the HCoV-host interactome aimed at the identification of candidate drugs with anti-SARS-CoV-2 effects.

The main and critical point that should be addressed is to provide a clear and adequate rationale for this study, highlighting the differences with the study that it is based on (Zhou et al, 2020). The authors have re-analyzed the HCoV-host interactome described in Zhou et al, and performed a different type of drug repurposing analysis whose results shared only one candidate drug with the original study.

More specifically:

1) Zhou et al generated their HCoV-host interactome based on validated published data. The authors of this novel manuscript found that 115 of the 135 proteins in the HCoV-host interactome showed no obvious interactions and focused their analysis on a set of 20 proteins forming a well-defined interactome. How are 115 proteins in the original HCoV-host interactome not connected? The authors should discuss their findings.

2) The added value and improvement over the original analysis is not clear, besides the addition of IL-6 (other cytokines might be considered as well). Zhou et al performed network-based drug repurposing and gene-set enrichment analysis. The authors of this manuscript should describe what weaknesses in the original work they address here with their analysis and compare and discuss the findings of the two analyses, highlighting the added value and improved accuracy of their study. Two almost completely different sets of drugs generated from the same data may raise more questions than provide answers. Thus, it is essential that the authors provide evidence that their analysis is, at the very least, significantly complementary to the original study.

3) What is the added value introduced by the integration of miRNA in the interaction map?

Two minor points:

1) Overall, the manuscript reads well but it should be carefully reviewed for syntax and incomplete sentences, e.g. “The resulting network is presented in” on Page 3 is incomplete. Similarly, the sentence “Concluding that IL-6 induces Bcl-2 expression to perform cytoprotective functions in response to oxygen toxicity, and that this effect is mediated by alterations in the interactions between BAK and MFNS” is incomplete as well.

2) The link http://precious.polito.it/covid-19 is broken.

**Have all data underlying the figures and results presented in the manuscript been provided?**

Reviewer #1: Yes

Reviewer #2: Yes

Reviewer #3: Yes

PLOS authors have the option to publish the peer review history of their article (what does this mean?). If published, this will include your full peer review and any attached files.

Reviewer #1: No

Reviewer #2: No

Reviewer #3: No
---

## [Decision Letter · Decision Letter 1]

11 Aug 2020

Dear Prof. Benso,

We are pleased to inform you that your manuscript 'IL6-mediated HCoV-host interactome regulatory network and GO/Pathway enrichment analysis' has been provisionally accepted for publication in PLOS Computational Biology.

Best regards,

James M. Briggs, Ph.D.

Associate Editor

PLOS Computational Biology

William Noble

Deputy Editor

PLOS Computational Biology

Reviewer's Responses to Questions

**Comments to the Authors:**

Reviewer #2: I am satisfied with the authors' responses to my comments. I still think that the computational approach to significantly speeding up hypothesis generation in a time of pandemics is the most valuable contribution here, and I think that the authors are still a bit too modest and underselling this point. That being said, in my opinion this manuscript is ready for publication.

**Have all data underlying the figures and results presented in the manuscript been provided?**

Reviewer #2: Yes

PLOS authors have the option to publish the peer review history of their article (what does this mean?). If published, this will include your full peer review and any attached files.

Reviewer #2: No

---

## [Editor Report · Acceptance letter]

24 Sep 2020

PCOMPBIOL-D-20-00555R1 

IL6-mediated HCoV-host interactome regulatory network and GO/Pathway enrichment analysis

Dear Dr Benso,

I am pleased to inform you that your manuscript has been formally accepted for publication in PLOS Computational Biology. Your manuscript is now with our production department and you will be notified of the publication date in due course.

With kind regards,

Matt Lyles
